# Exact conditions for evolutionary stability in indirect reciprocity under noise

**Nikoleta E. Glynatsi** [1,2]*, **Christian Hilbe**[3], **Yohsuke Murase**[1,2,4]

**1** RIKEN Center for Interdisciplinary Theoretical and Mathematical Science (iTHEMS), Wako, Japan,
**2** RIKEN Center for Computational Science, Kobe, Japan, **3** Interdisciplinary Transformation University,
Linz, Austria, **4** Graduate School of Science and Engineering, Saitama University, Saitama, Japan

* nikoleta.glynatsi@riken.jp

Exact conditions for evolutionary stability in
indirect reciprocity under noise. PLoS Comput
Biol 21(10): e1013584.

STATES OF AMERICA

**Peer Review History:** PLOS recognizes the
benefits of transparency in the peer review
process; therefore, we enable the publication of
all of the content of peer review and author
responses alongside final, published articles.
The editorial history of this article is available
here: https://doi.org/10.1371/journal.pcbi.
1013584

**Data availability statement:** The source code
used to numerically verify the results and

## Abstract

Indirect reciprocity is a key mechanism for large-scale cooperation. This mechanism
captures the insight that in part, people help others to build and maintain a good reputation. To enable such cooperation, appropriate social norms are essential. They specify
how individuals should act based on each others' reputations, and how reputations are
updated in response to individual actions. Although previous work has identified several
norms that sustain cooperation, a complete analytical characterization of all evolutionarily
stable norms remains lacking, especially when assessments or actions are noisy. In this
study, we provide such a characterization for the public assessment regime. This characterization reproduces known results, such as the leading eight norms, but it extends
to more general cases, allowing for various types of errors and additional actions including costly punishment. We also identify norms that impose a fixed payoff on any mutant
strategy, analogous to the zero-determinant strategies in direct reciprocity. These results
offer a rigorous foundation for understanding the evolution of cooperation through indirect
reciprocity and the critical role of social norms.

## Author summary

Understanding how cooperation can evolve and be sustained is a central question in evolutionary biology and social science. One prominent explanation is indirect reciprocity,
where individuals help others to build a good reputation and receive help in future.
For this mechanism to work, societies rely on social norms — shared rules that specify
how actions are judged and thereby how reputations are updated. Previous studies have
proposed specific norms that support cooperation. However, it has remained unclear
what general conditions make a norm evolutionarily stable. In this study, we develop a
mathematical framework to analytically derive such conditions. Our theory reproduces
well-known results, and it extends to more complex scenarios involving non-negligible
errors and costly punishment. These findings deepen our understanding of the evolution

generate the figures in this paper is available at: https://github.com/Nikoleta-v3/ ESS-conditions-indirect-reciprocity-with-noise.

**Funding:** Y.M. acknowledges support from the Japan Society for the Promotion of Science (JSPS) (JSPS KAKENHI; Grant no. JP25K07145). C.H. acknowledges generous funding from the European Research Council (ERC) under the European Union's Horizon 2020 research and innovation program (Starting Grant 850529: E-DIRECT). The funders had no role in study design, data collection and analysis, decision to publish, or preparation of the manuscript.

**Competing interests:** The authors have declared that no competing interests exist.

of cooperation and offer insights into how robust social norms can emerge and persist, even in noisy environments.

## 1. Introduction

Cooperation is a crucial aspect of life, and indirect reciprocity is a key mechanism to promote cooperation in human societies [1–6]. In indirect reciprocity, individuals decide how to treat others based on each other's reputations, and they cooperate to maintain a good reputation. Unlike direct reciprocity, in which individuals reciprocate based on their own experiences with others, indirect reciprocity relies on reputations as signals of past behavior. A concern for a good reputation may incentivize people to cooperate even with individuals they are unlikely to encounter again, enabling cooperation on a larger scale. To promote cooperation through indirect reciprocity, it is essential to have a proper "social norm". Such norms specify two components: an action rule, which prescribes how players should act based on others' reputations, and an assessment rule, which determines how reputations are updated in response to players' actions.

A major aim of this field is to identify evolutionarily stable norms, particularly those that promote cooperation. Previous studies have identified a number of such cooperative norms [7–21]. Among these, the so-called "leading eight" norms have received particular attention [9,10] (see Table 1 for their definition). The leading eight are fully cooperative, meaning that the population's cooperation rate approaches one when they are universally adopted. In addition, they are stable against invasion by any rare mutant strategy. They are characterized by four guiding principles: (i) Maintenance of cooperation: Good donors should cooperate with good recipients, and doing so should preserve their good reputation. (ii) Identification of defectors: Donors who defect against good recipients should be classified as bad. (iii) Justified punishment: Good donors may defect against bad recipients without harming their own reputation. (iv) Apology and forgiveness: Bad donors can restore their reputation by cooperating with good recipients. Overall, human behavior seems to be largely consistent with these principles, even though there is some mixed evidence on whether people regard justified punishment as truly justified [22–24].

To investigate cooperative norms, researchers have often focused on deterministic social norms, in which the assessment rule assigns reputations with certainty. Because the set of deterministic norms is finite, one can systematically enumerate all possibilities and identify

**Table 1. The prescriptions of the leading eight.** The top row $(X,Y)$ indicates the reputations of the donor and the recipient, respectively. For instance, $(G,B)$ refers to the case of a good $(G)$ donor who meets a bad $(B)$ recipient. The rules $S$, $R(C)$, $R(D)$ indicate the prescribed action, the assessment when cooperation $(C)$ is observed, and the assessment when defection $(D)$ is observed, respectively. An entry of 1 means the donor is assessed as good and 0 means the donor is assessed as bad. Those columns in which the leading eight differ from each other are highlighted in bold text.

| | $(G,G)$ | | | $(G,B)$ | | | $(B,G)$ | | | $(B,B)$ | | |
|---|---|---|---|---|---|---|---|---|---|---|---|---|
| | $S$ | $R(C)$ | $R(D)$ | $S$ | $R(C)$ | $R(D)$ | $S$ | $R(C)$ | $R(D)$ | $S$ | $R(C)$ | $R(D)$ |
| $L1$ (Standing) | C | 1 | 0 | D | **1** | 1 | C | 1 | 0 | **C** | **1** | **0** |
| $L2$ (Consistent Standing) | C | 1 | 0 | D | **0** | 1 | C | 1 | 0 | **C** | **1** | **0** |
| $L3$ (Simple Standing) | C | 1 | 0 | D | **1** | 1 | C | 1 | 0 | **D** | **1** | **1** |
| $L4$ | C | 1 | 0 | D | **1** | 1 | C | 1 | 0 | **D** | **0** | **1** |
| $L5$ | C | 1 | 0 | D | **0** | 1 | C | 1 | 0 | **D** | **1** | **1** |
| $L6$ (Stern Judging) | C | 1 | 0 | D | **0** | 1 | C | 1 | 0 | **D** | **0** | **1** |
| $L7$ (Staying) | C | 1 | 0 | D | **1** | 1 | C | 1 | 0 | **D** | **0** | **0** |
| $L8$ (Judging) | C | 1 | 0 | D | **0** | 1 | C | 1 | 0 | **D** | **0** | **0** |

those capable of sustaining cooperation under evolutionary pressure [7–21]. This approach can also incorporate nonzero error rates, allowing for occasional mistakes in actions or assessments. However, this enumerative method becomes infeasible for stochastic norms. In stochastic norms, the assessment rule may assign reputations probabilistically, leading to an uncountable number of possibilities [25,26]. To address this challenge, Murase et al. [26] derived exact analytical conditions for evolutionarily stable strategies (ESS) that sustain cooperation in the limit of vanishing error rates. Nevertheless, the current theory on stochastic norms remains limited to those that yield full cooperation in the vanishing-error limit. In this regime, the population converges to a homogeneous cooperative state in which all individuals are regarded as good and everyone cooperates. ESS conditions are then derived by analyzing whether rare deviations from this cooperative baseline can be profitable.

In this work, we remove these restrictions. Our methodological innovation is to calculate the long-term benefit of acquiring a good reputation, which in turn is the critical quantity needed to assess evolutionary stability. This quantity is relatively easy to derive under second-order social norms, where a donor's reputation does not persist beyond a single round, and it has been used to evaluate ESS [13,21,27]. Here, we extend the derivation to third-order norms. By evaluating whether maintaining a good reputation yields a higher long-term payoff than losing it, we can derive the necessary and sufficient conditions for all evolutionarily stable social norms, regardless of the cooperation level they sustain—an analysis that has been lacking. Importantly, our framework does not require errors to be vanishingly rare; it applies to arbitrary error rates. This generalization enables us to investigate more realistic scenarios, in which mistakes in assessment, action, or perception can occur. We further extend the framework to analyze additional actions beyond cooperation and defection, such as costly punishment [13,21]. Finally, we identify a novel class of norms that enforce a fixed payoff against any mutant strategy, reminiscent of zero-determinant strategies in direct reciprocity [28].

The paper is organized as follows. In Sect 2, we introduce the model and establish useful notation. Sect 3 develops our analytical framework and shows how to calculate the long-term benefit of acquiring a good reputation. Using this framework, we obtain the following main results: First, we derive necessary and sufficient conditions for the evolutionary stability of third-order norms under various types of errors at arbitrary rates. Second, we extend the framework to incorporate additional actions, focusing on costly punishment. Third, we apply our results to investigate several special cases (Sect 4): (i) cooperative ESS in the limit of vanishing errors, (ii) cooperative ESS with costly punishment in the limit of vanishing errors, (iii) stability of the leading eight norms in the presence of various errors, and (iv) finally, we characterize a novel class of norms, the "equalizer" norms, which enforce a fixed payoff against any mutant strategy. The last section summarizes our findings and discusses their implications.

## 2. Model

In this study, we follow the basic framework of Ohtsuki and Iwasa [9]. We consider an infinitely large population of players who interact in pairwise donation games. In each round, two players are randomly chosen as a donor and a recipient, respectively. The donor decides whether to cooperate ($C$) or to defect ($D$). Cooperation incurs a cost $c>0$ for the donor and results in a benefit $b>c$ for the recipient. Defection leads to a payoff of zero for both players. If the donation game is only played once, the donor is better off by defecting, creating a social dilemma. However, here we assume that population members play many donation games,

against different opponents. In that case, their actions can affect their reputation, which in turn may influence how they are treated in future.

We assume reputations are binary and public. That is, the reputation of a player can be either good ($G$) or bad ($B$), and it is known to all other players without any disagreement. How players form reputations, and how they act based on these reputations, depends on their social norm. In our study, a social norm consists of an action rule and an assessment rule, as shown in Fig 1.

Social norms are often categorized by their order, which reflects the information on which actions and assessments are based. First-order norms assess the donor's reputation based solely on the donor's action, without considering the context or the recipient's reputation. Second-order norms take into account both the donor's action and the recipient's reputation, enabling distinctions such as justified vs. unjustified defection. The action rule depends only on the donor's reputation in first- and second-order norms. Third-order norms additionally consider the donor's own reputation, allowing for more nuanced assessments. Assessment rules and action rules in third-order norms can depend on the reputations of both the donor and the recipient. Following [26], we consider a stochastic version of third-order social norms in this paper.

A social norm's action rule $S(X,Y)$ determines which action a player takes as a donor. This choice might depend on the player's own reputation $X$ as well as on the reputation $Y$ of the recipient, where $X,Y \in \{G,B\}$. The output $S(X,Y) \in \{C,D\}$ is the action that the donor takes. Here, we assume that the action rule is deterministic (that is, donors cooperate with probability zero or one). This assumption is without loss of generality, since stochastic action rules cannot be evolutionarily stable [26]: For a given context, the best response is uniquely determined except for the special cases where the expected payoffs of the two actions are equal (in which case neutral drift would be possible). In the following we exclude those special cases from our analysis.

A social norm's assessment rule $R(X,Y,A)$ determines the probability that the donor is assigned a good reputation after the interaction. This probability depends on the previous reputation $X$ of the donor, the previous reputation $Y$ of the recipient, and on the donor's action $A \in \{C,D\}$ in the donation game. When the output of an assessment rule $R(X,Y,A)$ is constrained to be either zero or one for any input $(X,Y,A)$, the norm is deterministic; otherwise it is stochastic.

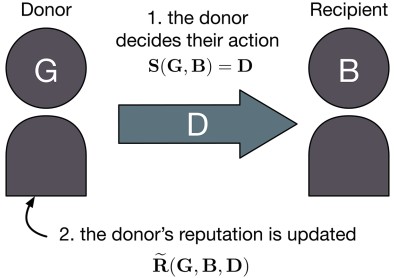

**Fig 1. Schematic representation of the model.** At each time step, two players are randomly chosen, one as the donor and the other as the recipient. The donor chooses an action according to the action rule $S(X,Y)$, which depends on the reputation of the donor $X$ and the reputation of the recipient $Y$. After the interaction, the assessment rule $R(X,Y,A)$ determines the donor's new reputation. This reputation depends on the donor's previous reputation $X$, the recipient's previous reputation $Y$, and the donor's action $A$. The donor is assigned a good reputation with probability $\widetilde{R}(X,Y,A)$, which is the effective assessment rule that accounts for errors in assessment. We repeat this process indefinitely, and we are interested in the population's long-term behavior.

We introduce assessment errors, which occur when new reputations are assigned. With probabilities $\mu$, the respective assignments are the opposite of the assignment prescribed by the social norm. As a result, instead of their intended assessment rules, players implement the effective assessment rules

$$\widetilde{R}(X, Y, A) = (1 - \mu) R(X, Y, A) + \mu [1 - R(X, Y, A)]. \tag{1}$$

In the presence of these errors, we obtain the constraint $\mu \leq \widetilde{R}(X, Y, A) \leq 1 - \mu$. When $\mu > 0$, the reputation dynamics are ergodic. This means that over time, the system explores all possible reputation states and that its long-term behavior becomes independent of the initial reputation configuration [26].

In agreement with the seminal work of Ohtsuki and Iwasa [9,10], we consider a public assessment model. That is, all players learn the same information and share the same assessment of any given population member at any point in time. These shared assessments can change in time, depending on the population members' interactions. Herein, we assume players interact in sufficiently many donation games such that their reputation assignments reach a stationary state.

In the remainder of this article, we focus on identifying which social norms are ESS. We refer to the norm adopted by the majority of the population as the resident norm. For positive error rates, we require the resident norm to form a strict Nash equilibrium: if an infinitesimal minority of the population adopts a different norm, the minority receives a strictly lower payoff than the residents. Because in the public assessment model the reputation-updating mechanism is externally defined and shared at the population level, individual mutants cannot change it. It is therefore sufficient to consider mutants with different action rules but identical assessment rules as the resident. Note that under this framework, at most two different norms can be present at any time. Thus, we do not consider scenarios in which multiple action rules coexist simultaneously [29].

We also focus on the particularly important special case, already discussed, of social norms that are not only ESS but also *self-cooperative*. When a self-cooperative norm is adopted by everyone, the population's cooperation rate approaches one in the limit of rare errors. We refer to such norms as cooperative ESS (CESS) [10,14,26].

## 3. Results

To characterize all ESS, we first describe how reputations evolve over time. As a crucial measure, we obtain the equilibrium fraction of good players in the population (Sect 3.1). Using this equilibrium fraction, we calculate the long-term benefit of acquiring a good reputation (Sect 3.2). Based on these results, we derive necessary and sufficient conditions for a social norm to be an ESS (Sect 3.3). These conditions are then naturally extended to account for other types of errors (Sect 3.4) and for additional actions (Sect 3.5).

### 3.1. Description of the reputation dynamics

Consider a homogeneous population with action rule $S(X,Y)$ and assessment rule $R(X,Y,A)$, which together define the resident norm. At any given time $t$, let $h(t)$ denote the fraction of players with a good reputation. Similarly, $1 - h(t)$ is the fraction of players with a bad reputation. Then $h(t)$ obeys the following differential equation,

$$
\begin{aligned}
\dot{h}(t) \;=\;& h(t)^2 R_S(G,G) \\
+\;& h(t)\left(1-h(t)\right)\left[R_S(G,B)+R_S(B,G)\right] \\
+\;& \left(1-h(t)\right)^2 R_S(B,B) \\
-\;& h(t).
\end{aligned}
\tag{2}
$$

In this expression, $R_S(X,Y)$ is the probability to assign a good reputation to the donor if the donor's and recipient's initial reputations are $X$ and $Y$, respectively. This probability is defined as

$$
R_S(X,Y) \equiv \widetilde{R}(X,Y,S(X,Y)).
\tag{3}
$$

As $t \to \infty$, the proportion of good population members $h(t)$ converges to a fixed point $h^* \in [0,1]$. This fixed point is unique and stable, because the above equation is quadratic with respect to $h$ and because $\dot{h}|_{h=1} < 0$ and $\dot{h}|_{h=0} > 0$ when $\mu_a > 0$. By plugging $\dot{h} = 0$ into Eq (2), the stationary value is obtained as a solution to the quadratic equation

$$
c_2 h^{*2} + c_1 h^* + c_0 = 0,
\tag{4}
$$

where $c_2$, $c_1$, and $c_0$ are defined as

$$
\begin{aligned}
c_2 &\equiv R_S(G,G) - R_S(G,B) - R_S(B,G) + R_S(B,B) \\
c_1 &\equiv R_S(G,B) + R_S(B,G) - 2R_S(B,B) - 1 \qquad . \\
c_0 &\equiv R_S(B,B)
\end{aligned}
\tag{5}
$$

The unique solution $h^* \in [0,1]$ to the quadratic equation (4) is

$$
h^* =
\begin{cases}
\dfrac{-c_1 - \sqrt{c_1^2 - 4c_2 c_0}}{2c_2} & \text{when } c_2 \neq 0 \\[2ex]
-\dfrac{c_0}{c_1} & \text{when } c_2 = 0.
\end{cases}
\tag{6}
$$

(The other solution to the quadratic equation is not in the unit interval [0,1]).

At the stationary state, the probability that a donor takes action $A \in \{C,D\}$ when interacting with another member of the population is

$$
p_A^{\text{res}\to\text{res}} = h^{*2}\chi_A(G,G) + h^*(1-h^*)\left[\chi_A(G,B)+\chi_A(B,G)\right] + (1-h^*)^2\chi_A(B,B).
\tag{7}
$$

Here, "res" refers to an individual following the resident norm, and the arrow denotes *donor → recipient*. Thus, $p_A^{\text{res}\to\text{res}}$ is the probability that a resident donor takes action $A$ toward a resident recipient. Moreover, $\chi_A$ is an indicator function defined by:

$$
\chi_A(X,Y) \equiv
\begin{cases}
1 & \text{if } S(X,Y) = A \\
0 & \text{otherwise.}
\end{cases}
\tag{8}
$$

In particular, for the social norm to be self-cooperative, $p_C^{\text{res}\to\text{res}}$ must converge to one as $\mu \to 0$.

### 3.2. Long-term benefit of having a good reputation

In the following, we derive a necessary and sufficient condition for a social norm to be an ESS. To this end, we first calculate the expected long-term payoff of a player who is currently assigned a good or a bad reputation, respectively. We use this expression to check if the social norm's action rule is the unique best response in all possible contexts. Here, the possible contexts refer to all possible combinations of the donor's and the recipient's reputations, $(G,G)$, $(B,G)$, $(G,B)$, and $(B,B)$.

Suppose there is a good player following the social norm $(R,S)$. We consider the player's cumulative payoff for the subsequent $T$ rounds,

$$v_G^{(T)} \equiv \sum_{t=1}^{T} \langle \pi_G^{(t)} \rangle. \tag{9}$$

Here, $\langle \pi_G^{(t)} \rangle$ is the expected payoff in the $t$-th round, given the player initially has a $G$ reputation. A round is defined as a single donation game, in which the player is the donor or the recipient, each with probability 1/2. The cumulative payoff $v_B^{(T)}$ for a $B$ player is defined analogously.

To derive an explicit expression for the cumulative payoff $v_G^{(t)}$, consider a focal player with an initially good reputation. We distinguish two possible cases that could occur in the player's next game. ($i$) If the player happens to act as the recipient in the next game, this player receives a benefit $b$ with probability $h^* \chi_C(G, G) + (1 - h^*) \chi_C(B, G)$, because the donor is $G$ with probability $h^*$ and $B$ with probability $1 - h^*$. In that case, the player maintains their previous reputation. ($ii$) Alternatively, if the player acts as the donor, this player pays the cost $c$ with probability $h^* \chi_C(G, G) + (1 - h^*) \chi_C(G, B)$. Now, the player's reputation is updated according to the assessment rule $R$. The donor is assigned a good reputation with probability $R_S(G, G)$ if they met a $G$ recipient, and with probability $R_S(G, B)$ if they met a $B$ recipient. If they obtain a good reputation, they obtain the payoff $v_G^{(T-1)}$ in the subsequent $T$–1 rounds. If they obtain a bad reputation, their subsequent payoff is $v_B^{(T-1)}$. Overall, the expected cumulative payoff of a $G$ player is

$$
\begin{aligned}
v_G^{(T)} = \frac{1}{2} \cdot & \Big[ b \left[ h^* \chi_C(G, G) + (1 - h^*) \chi_C(B, G) \right] + v_G^{(T-1)} \Big] \\
+ \frac{1}{2} \cdot & \Big[ - c \left[ h^* \chi_C(G, G) + (1 - h^*) \chi_C(G, B) \right] \\
& + h^* R_S(G, G) v_G^{(T-1)} + (1 - h^*) R_S(G, B) v_G^{(T-1)} \\
& + h^* \left[ 1 - R_S(G, G) \right] v_B^{(T-1)} + (1 - h^*) \left[ 1 - R_S(G, B) \right] v_B^{(T-1)} \Big].
\end{aligned}
\tag{10}
$$

Similarly, the expected payoff of a $B$ player in the subsequent $T$ rounds is

$$
\begin{aligned}
v_B^{(T)} = \frac{1}{2} \cdot & \Big[ b \left[ h^* \chi_C(G, B) + (1 - h^*) \chi_C(B, B) \right] + v_B^{(T-1)} \Big] \\
+ \frac{1}{2} \cdot & \Big[ - c \left[ h^* \chi_C(B, G) + (1 - h^*) \chi_C(B, B) \right] \\
& + h^* R_S(B, G) v_G^{(T-1)} + (1 - h^*) R_S(B, B) v_G^{(T-1)} \\
& + h^* \left[ 1 - R_S(B, G) \right] v_B^{(T-1)} + (1 - h^*) \left[ 1 - R_S(B, B) \right] v_B^{(T-1)} \Big].
\end{aligned}
\tag{11}
$$

The difference between these two expected payoffs is

$$
\begin{aligned}
v_G^{(T)} - v_B^{(T)} = \frac{1}{2}\Bigg[ & b\left[h^* \chi_C(G,\Delta) + (1-h^*)\chi_C(B,\Delta)\right] \\
& - c\left[h^* \chi_C(\Delta,G) + (1-h^*)\chi_C(\Delta,B)\right] \\
& + \left(v_G^{(T-1)} - v_B^{(T-1)}\right)\left\{1 + h^* R_S(\Delta,G) + (1-h^*)R_S(\Delta,B)\right\}\Bigg].
\end{aligned}
\tag{12}
$$

Here, we use the following definitions for $X \in \{G,B\}$

$$
\begin{aligned}
\chi_C(X,\Delta) &\equiv \chi_C(X,G) - \chi_C(X,B), \\
\chi_C(\Delta,X) &\equiv \chi_C(G,X) - \chi_C(B,X), \\
R_S(\Delta,X) &\equiv R_S(G,X) - R_S(B,X).
\end{aligned}
\tag{13}
$$

As we saw in the previous section, the system converges to a stationary state where the fraction of good players is $h^*$ irrespective of the initial reputation configuration. Therefore, the expected payoffs in the $t$-th round, $\langle \pi_G^{(t)} \rangle$ and $\langle \pi_B^{(t)} \rangle$, converge to the same value in the limit as $t \to \infty$. Hence, the difference $v_G^{(T)} - v_B^{(T)}$ approaches a constant value as $T$ becomes large. Let us define the respective limit as

$$
\Delta v \equiv \lim_{T \to \infty}\left(v_G^{(T)} - v_B^{(T)}\right).
\tag{14}
$$

We can obtain an implicit equation for $\Delta v$ by taking the limit $T \to \infty$ in Eq (12). By solving the resulting expression for $\Delta v$, we obtain

$$
\Delta v = \frac{b\left[h^* \chi_C(G,\Delta) + (1-h^*)\chi_C(B,\Delta)\right] - c\left[h^* \chi_C(\Delta,G) + (1-h^*)\chi_C(\Delta,B)\right]}{1 - h^* R_S(\Delta,G) - (1-h^*)R_S(\Delta,B)}.
\tag{15}
$$

The first term in the numerator can be interpreted as the expected benefit a $G$ player obtains compared to a $B$ player. The second term is the expected cost a $G$ player additionally pays compared to a $B$ player. The denominator indicates how long the initial reputation lasts. When it takes more time steps to recover from a bad reputation, $R_S(\Delta,G)$ and $R_S(\Delta,B)$ tend to be larger. With such a "sticky" social norm, the denominator becomes smaller and $\Delta v$ becomes larger. In other words, being assessed as $G$ has a larger impact on the player's long-term payoff.

The expression simplifies considerably for second-order norms. In these norms, neither the action nor the assessment depends on the donor's reputation. As a result, $R_S(\Delta,G) = R_S(\Delta,B) = 0$ and $\chi_C(\Delta,G) = \chi_C(\Delta,B) = 0$ hold. If we further assume a discriminating action rule, which prescribes cooperation for good recipients and defection for bad recipients, then $\chi_C(G,\Delta) = \chi_C(B,\Delta) = 1$. In that case, Eq (15) reduces to the simple form

$$
\Delta v = b.
\tag{16}
$$

That is, under such a norm, the long-run advantage of a good reputation is equivalent to receiving an additional benefit $b$ in one round.

### 3.3. ESS conditions

A social norm is an ESS if and only if the resident action rule is the best response in all possible contexts, $(G,G)$, $(B,G)$, $(G,B)$, and $(B,B)$.

First, let us consider the context $(G,G)$ as an example. For $S(G,G) = C$ to be the best response, the following condition must hold:

$$-c + \widetilde{R}(G,G,C)\,\Delta v > \widetilde{R}(G,G,D)\,\Delta v. \tag{17}$$

The left-hand side of the equation is the expected payoff of a $G$ player when they cooperate, and the right-hand side is the expected payoff when they defect. The equation can be simplified as follows,

$$\left[\widetilde{R}(G,G,C) - \widetilde{R}(G,G,D)\right]\Delta v > c. \tag{18}$$

The left-hand side of the equation is the expected long-term benefit of having a good reputation while the right-hand side is the immediate cost of cooperation. If this inequality holds, $S(G,G) = C$ is the best response. Conversely, if the inequality is reversed, $S(G,G) = D$ is the best response. Similarly, we can analyze the other possible contexts. As a result, we obtain the following characterization of ESS norms.

**Theorem 1.** *A third-order social norm with assessment rule R(X,Y,A) and action rule S(X,Y) is an ESS if and only if*

$$\begin{cases} \left[\widetilde{R}(X,Y,C) - \widetilde{R}(X,Y,D)\right]\Delta v > c & \text{if } S(X,Y) = C \\ \left[\widetilde{R}(X,Y,C) - \widetilde{R}(X,Y,D)\right]\Delta v < c & \text{if } S(X,Y) = D \end{cases} \tag{19}$$

*holds for all possible contexts* $(X,Y) \in \left\{(G,G),(G,B),(B,G),(B,B)\right\}$.

Consider ALLD (Always Defect: $S(*,G) = S(*,B) = D$) as an example. Under this norm, $\Delta v = 0$ because $\chi_C(G,\Delta) = \chi_C(B,\Delta) = \chi_C(\Delta,G) = \chi_C(\Delta,B) = 0$. As a result, Eq (19) is satisfied for all contexts $(X,Y)$ because the left-hand side evaluates to zero.

### 3.4. ESS conditions with perception and implementation errors

So far, we have considered only assessment errors. In the following, we show how the respective results can be applied to other types of errors, by rescaling the effective assessment rules and the effective benefit and cost of cooperation.

First, we consider the case of misperception errors. Specifically, we assume that when a player defects, the action is mistakenly perceived as cooperation with probability $\epsilon_{DC}$ (it is correctly perceived as defection with probability $1 - \epsilon_{DC}$). This assumption may reflect, for example, that defectors have a natural incentive to deceive bystanders and to misrepresent their actions. In this case of such misperception errors, the effective assessment rule becomes

$$\begin{aligned} \widetilde{R}(X,Y,C)^* &\equiv \widetilde{R}(X,Y,C) \\ \widetilde{R}(X,Y,D)^* &\equiv (1 - \epsilon_{DC})\,\widetilde{R}(X,Y,D) + \epsilon_{DC}\widetilde{R}(X,Y,C), \end{aligned} \tag{20}$$

for any $X, Y \in \{G, B\}$. The ESS conditions for the case with the perception error is the same as Eq (19), but now with the rescaled assessment rules. Similarly, we could also

consider other types of perception errors, such as the case where cooperations are misperceived as defections.

Implementation errors represent another type of error that is frequently studied in the literature. When actions are subject to implementation errors, individuals who intend to cooperate may sometimes defect, for example because of a lack of resources. Let $\mu_e$ be the respective (implementation) error rate. Note that here, we assume that defections are always implemented perfectly, without errors. In the presence of such implementation errors, the cooperation probabilities $\chi_C(X, Y)$ are rescaled as $(1 - \mu_e) \chi_C(X, Y)$. In the above analysis, this rescaling in $\chi_C(X, Y)$ is equivalent to the rescaling of the effective assessment rules and the effective benefit and cost of cooperation,

$$
\begin{aligned}
\widetilde{R}(X, Y, C)^{\ddagger} &\equiv (1 - \mu_e)\, \widetilde{R}(X, Y, C)^* + \mu_e \widetilde{R}(X, Y, D)^* \\
\widetilde{R}(X, Y, D)^{\ddagger} &\equiv \widetilde{R}(X, Y, D)^* \\
b^{\ddagger} &\equiv (1 - \mu_e)\, b \\
c^{\ddagger} &\equiv (1 - \mu_e)\, c,
\end{aligned}
\tag{21}
$$

Here, the effective assessment rule $\widetilde{R}(X, Y, C)^{\ddagger}$ indicates the probability that an $X$-donor is assigned a good reputation, given they intended to cooperate with $Y$. In that case, the donor pays the effective cost while the recipient receives the effective benefit. The ESS condition for the case with the implementation error is the same as Eq (19) but with the rescaled parameters,

$$
\begin{cases}
\left[ \widetilde{R}(X, Y, C)^{\ddagger} - \widetilde{R}(X, Y, D)^{\ddagger} \right] \Delta v^{\ddagger} > c^{\ddagger} & \text{if } S(X, Y) = C, \\
\left[ \widetilde{R}(X, Y, C)^{\ddagger} - \widetilde{R}(X, Y, D)^{\ddagger} \right] \Delta v^{\ddagger} < c^{\ddagger} & \text{if } S(X, Y) = D.
\end{cases}
\tag{22}
$$

Here, $\Delta v^{\ddagger}$ is $\Delta v$ in Eq (15) with the appropriately rescaled parameters.

To gain insights into the effect of these errors, let us consider the L6 norm (Stern Judging) as an example. According to Eq (22), L6 is an ESS if and only if

$$
\frac{b}{c} > \frac{1}{(1 - \epsilon_{DC})(1 - \mu_e)(1 - 2\mu)}.
\tag{23}
$$

As the error rates $\mu$, $\mu_e$, and $\epsilon_{DC}$ increase, the lower bound of $b/c$ diverges and cooperation gets harder to maintain. This reproduces the results in [21].

### 3.5. ESS conditions when other actions are available

We can also extend the above analysis to the case where additional actions are available. As an example, we consider the case that a player can exert costly punishment ($P$). In that case, the donor reduces the recipient's payoff by $\beta > 0$, at an own cost of $\alpha > 0$. The resulting dynamics of $h^*$ remains the same as Eq (2) and the solution for $h^*$ is the same as Eq (6). The analysis in Sect 3.1 is also valid for the case with punishment, except that now we need to consider the additional action $P$. The expected payoff of a $G$ player in the subsequent $T$ rounds becomes

$$v_G^{(T)} = \frac{1}{2} \cdot \Big[ b \left[ h^* \chi_C (G, G) + (1 - h^*) \chi_C (B, G) \right]$$

$$- \beta \left[ h^* \chi_P (G, G) + (1 - h^*) \chi_P (B, G) \right] + v_G^{(T-1)} \Big]$$

$$+ \frac{1}{2} \cdot \Big[ (-c) \left[ h^* \chi_C (G, G) + (1 - h^*) \chi_C (G, B) \right]$$

$$- \alpha \left[ h^* \chi_P (G, G) + (1 - h^*) \chi_P (G, B) \right] \tag{24}$$

$$+ h^* R_S (G, G) v_G^{(T-1)} + (1 - h^*) R_S (G, B) v_G^{(T-1)}$$

$$+ h^* \left[ 1 - R_S (G, G) \right] v_B^{(T-1)} + (1 - h^*) \left[ 1 - R_S (G, B) \right] v_B^{(T-1)} \Big],$$

where $\chi_P (X, Y)$ is the punishing probability, defined analogously to Eq (8). The difference between the expected payoffs of a $G$ and a $B$ player is now

$$v_G^{(T)} - v_B^{(T)} = \frac{1}{2} \cdot \Bigg[ b \left[ h^* \chi_C (G, \Delta) + (1 - h^*) \chi_C (B, \Delta) \right]$$

$$- c \left[ h^* \chi_C (\Delta, G) + (1 - h^*) \chi_C (\Delta, B) \right]$$

$$- \alpha \left[ h^* \chi_P (G, \Delta) + (1 - h^*) \chi_P (B, \Delta) \right] \tag{25}$$

$$- \beta \left[ h^* \chi_P (\Delta, G) + (1 - h^*) \chi_P (\Delta, B) \right]$$

$$+ \left( v_G^{(T-1)} - v_B^{(T-1)} \right) \left\{ 1 + h^* R_S (\Delta, G) + (1 - h^*) R_S (\Delta, B) \right\} \Bigg],$$

where $\chi_P (X, \Delta)$ and $\chi_P (\Delta, X)$ are defined analogously to $\chi_C (X, \Delta)$ and $\chi_C (\Delta, X)$, respectively. The expected payoff difference $\Delta v$ is obtained by taking the limit of $T \to \infty$ in Eq (25).

$$\Delta v = \frac{b \overline{\chi_C} (h^*, \Delta) - c \overline{\chi_C} (\Delta, h^*) - \beta \overline{\chi_P} (h^*, \Delta) - \alpha \overline{\chi_P} (\Delta, h^*)}{1 - h^* R_S (\Delta, G) - (1 - h^*) R_S (\Delta, B)}, \tag{26}$$

where we defined

$$\overline{\chi_C} (h^*, \Delta) \equiv h^* \chi_C (G, \Delta) + (1 - h^*) \chi_C (B, \Delta)$$

$$\overline{\chi_C} (\Delta, h^*) \equiv h^* \chi_C (\Delta, G) + (1 - h^*) \chi_C (\Delta, B)$$

$$\overline{\chi_P} (h^*, \Delta) \equiv h^* \chi_P (G, \Delta) + (1 - h^*) \chi_P (B, \Delta) \tag{27}$$

$$\overline{\chi_P} (\Delta, h^*) \equiv h^* \chi_P (\Delta, G) + (1 - h^*) \chi_P (\Delta, B).$$

Using $\Delta v$, we can derive the ESS conditions for norms with punishment. The action prescribed by the social norm is the unique best response for context $(X, Y)$ if and only if both other actions yield lower payoffs. For instance, the action rule $S(X, Y) = C$ is the best response if and only if

$$\left[ \widetilde{R} (X, Y, C) - \widetilde{R} (X, Y, D) \right] \Delta v > c \quad \text{and} \quad \left[ \widetilde{R} (X, Y, C) - \widetilde{R} (X, Y, P) \right] \Delta v > c - \alpha. \tag{28}$$

Similarly, the action rule $S(X, Y) = D$ is the best response if and only if

$$\left[ \widetilde{R} (X, Y, D) - \widetilde{R} (X, Y, C) \right] \Delta v > -c \quad \text{and} \quad \left[ \widetilde{R} (X, Y, D) - \widetilde{R} (X, Y, P) \right] \Delta v > -\alpha. \tag{29}$$

Finally, the action rule $S(X, Y) = P$ is the best response if and only if

$$\left[\widetilde{R}(X, Y, P) - \widetilde{R}(X, Y, C)\right]\Delta v > \alpha - c \quad \text{and} \quad \left[\widetilde{R}(X, Y, P) - \widetilde{R}(X, Y, D)\right]\Delta v > \alpha. \tag{30}$$

The social norm is an ESS if and only if the above conditions hold for all contexts $(X, Y) \in \{(G, G), (G, B), (B, G), (B, B)\}$. It is straightforward to generalize the above analysis to the case where further actions are available.

## 4. Special cases

To illustrate the scope and power of our analytical framework, we next apply it to several special cases that have been central to the literature on indirect reciprocity. First, we characterize cooperative ESS in the limit of vanishing errors, showing how our framework recovers previous results (Sect 4.1). Second, we analyze the role of costly punishment in promoting cooperation (Sect 4.2). Third, we study the stability of the leading eight norms in the presence of errors (Sect 4.3). Finally, we identify a novel class of "equalizer" norms that enforce fixed payoffs against any mutant strategy (Sect 4.4).

### 4.1. Self-cooperative ESS in the limit of vanishing error rates

In this section, we focus on cooperative ESS (CESS), which are a special subset of the ESS norms. A norm is a CESS if it satisfies the following two conditions in the limit of vanishing error rates,

(a) The social norm is fully self-cooperative, i.e., $p_C^{\text{res}\to\text{res}} \to 1$ as $\mu \to 0^+$.
(b) The social norm is an ESS.

In the following, the effective assessment rule converges to the original assessment rule, $\widetilde{R}(X, Y, A) \to R(X, Y, A)$, as $\mu \to 0^+$.

First, we show that for any such CESS, either $h^* = 1$ or $h^* = 0$ must hold. Assume to the contrary that $0 < h^* < 1$, such that there are both good and bad players in the population. For the norm to be self-cooperative, the action rule then needs to prescribe cooperation in all possible cases. The resulting norm of unconditional cooperation, however, is not an ESS. As the two labels $G$ and $B$ are interchangeable [9], we consider without loss of generality the case that $h^* = 1$ in the following. When the respective social norm is adopted by the entire population, we assume everyone is assigned a good reputation eventually.

First, we check the condition (a). To have $h^* = 1$, the following conditions are necessary and sufficient:

$$h^* = 1 \iff \begin{cases} \dot{h}|_{h=1} = 0 \\ \frac{d\dot{h}}{dh}\Big|_{h=1} < 0 \end{cases}. \tag{31}$$

The first equation on the right hand side makes sure there is a fixed point at $h = 1$. The second inequality indicates that this fixed point is stable. By Eq (2) these two requirements are equivalent to the following conditions,

$$\begin{cases} R_S(G, G) = 1 \\ R_S(G, B) + R_S(B, G) > 1. \end{cases} \tag{32}$$

Given these conditions are satisfied, the social norm is self-cooperative if and only if

$$S(G, G) = C. \tag{33}$$

We conclude that the self-cooperative norms in which all population members have a good reputation are exactly those that satisfy conditions (32) and (33). For self-cooperative norms, $h^* = 1$, Eq (15) simplifies to

$$\Delta v = \frac{b\chi_C(G, \Delta) - c\chi_C(\Delta, G)}{1 - R_S(\Delta, G)}. \tag{34}$$

Second, we check the ESS condition (b). To this end, we use Eq (19) for the contexts $(G,G)$, $(G,B)$, $(B,G)$, and $(B,B)$ in the following.

1. For the context $(X, Y) = (G, G)$, the ESS condition (19) is

$$[R(G, G, C) - R(G, G, D)]\Delta v > c$$
$$[R(G, G, C) - R(G, G, D)]\{b[1 - \chi_C(G, B)] - c[1 - \chi_C(B, G)]\} > cR_S(B, G), \tag{35}$$

   where we used Eqs (32) and (33) for the derivation of the second line. For this inequality to hold, $\chi_C(G, B) = 0$ is necessary. Thus,

$$[R(G, G, C) - R(G, G, D)]\{b - c[1 - \chi_C(B, G)]\} > cR_S(B, G). \tag{36}$$

2. For the context $(X, Y) = (G, B)$ we have shown previously that the action rule must prescribe $S(G, B) = D$. This is the best response if and only if

$$[R(G, B, D) - R(G, B, C)]\Delta v > -c$$
$$[R(G, B, D) - R(G, B, C)]\{b - c[1 - \chi_C(B, G)]\} > -cR_S(B, G). \tag{37}$$

3. For the context $(X, Y) = (B, G)$, the action rule may be either $S(B, G) = C$ or $S(B, G) = D$. When $S(B, G) = C$, the best response condition is

$$[R(B, G, C) - R(B, G, D)]\Delta v > c$$
$$[R(B, G, C) - R(B, G, D)]\{b - c[1 - \chi_C(B, G)]\} > cR_S(B, G). \tag{38}$$

   When $S(B, G) = D$, the best response condition is

$$[R(B, G, D) - R(B, G, C)]\Delta v > -c$$
$$[R(B, G, D) - R(B, G, C)]\{b - c[1 - \chi_C(B, G)]\} > -cR_S(B, G). \tag{39}$$

4. Finally, for the context $(X, Y) = (B, B)$, the action rule $S(B, B) = C$ is the best response if

$$[R(B, B, C) - R(B, B, D)]\Delta v > c$$
$$[R(B, B, C) - R(B, B, D)]\{b - c[1 - \chi_C(B, G)]\} > cR_S(B, G). \tag{40}$$

   When the inequality is reversed, $S(B, B) = D$ is the best response.

To summarize, a social norm constitutes a CESS if and only if one of two conditions is satisfied. These conditions are distinguished based on the value of $S(B,G)$, that is, based on the action of a bad donor who encounters a good recipient.

When $S(B, G) = C$, the CESS condition is:

$$\begin{cases} S(G, G) = C \\ S(G, B) = D \\ S(B, G) = C \\ R(G, G, C) = 1 \\ R(G, B, D) + R(B, G, C) > 1 \\ [R(G, G, C) - R(G, G, D)]\, b > cR(B, G, C) \\ [R(G, B, C) - R(G, B, D)]\, b < cR(B, G, C) \\ [R(B, G, C) - R(B, G, D)]\, b > cR(B, G, C) \\ S(B, B) = \begin{cases} C & \text{if } [R(B, B, C) - R(B, B, D)]\, b > cR(B, G, C) \\ D & \text{if } [R(B, B, C) - R(B, B, D)]\, b < cR(B, G, C) \end{cases} \end{cases} \qquad (41)$$

If the assessment rule is additionally assumed to be deterministic, this set of conditions reproduces the leading-eight social norms, as shown in the top row of Table 2. They are stable for $b > c$ [26].

When $S(B, G) = D$, the CESS condition is:

$$\begin{cases} S(G, G) = C \\ S(G, B) = D \\ S(B, G) = D \\ R(G, G, C) = 1 \\ R(G, B, D) + R(B, G, D) > 1 \\ [R(G, G, C) - R(G, G, D)]\,(b - c) > cR(B, G, D) \\ [R(G, B, C) - R(G, B, D)]\,(b - c) < cR(B, G, D) \\ [R(B, G, C) - R(B, G, D)]\,(b - c) < cR(B, G, D) \\ S(B, B) = \begin{cases} C & \text{if } [R(B, B, C) - R(B, B, D)]\,(b - c) > cR(B, G, D) \\ D & \text{if } [R(B, B, C) - R(B, B, D)]\,(b - c) < cR(B, G, D) \end{cases} \end{cases} \qquad (42)$$

If the norm is deterministic, we recover the secondary-sixteen social norms, see the bottom row of Table 2 [26]. They are stable for $b>2c$ [26]. The leading eight and the secondary sixteen are the only CESS when assessment rules are deterministic. In contrast, in the stochastic case there exists a spectrum of CESS, characterized by the conditions (41) and (42).

## 4.2. Self-cooperative ESS norms with punishment

Next we consider the CESS norms when punishment is available. Suppose the action rule is

$$\begin{aligned} S(G, G) &= C \\ S(G, B) &= A_{GB} \in \{D, P\} \\ S(B, G) &= A_{BG} \in \{C, D, P\} \\ S(B, B) &= A_{BB} \in \{C, D, P\}, \end{aligned} \qquad (43)$$

**Table 2**. **Deterministic CESS in the limit of vanishing errors.** The CESS can be categorized into two classes, the leading-eight norms (top) and the secondary-sixteen norms (bottom), respectively. In this table, the left most column indicates the original reputations of the donor and the recipient. The second column then shows the norm's action rule and the third and fourth column its assessment rule. The rightmost column gives the condition for the norm to be a CESS. The symbol * indicates that the respective value can be either 0 or 1. In this table, the assessment rule for context $(B,B)$ is not shown as it can be arbitrary. Given the respective entries $R(B,B,*)$ and the environmental conditions $\{b, c\}$, the optimal action $S(B,B)$ is uniquely determined.

| (Donor rep, Recipient rep) $(X,Y)$ | Action rule $S(X,Y)$ | Reputation update based on action | | condition |
|---|---|---|---|---|
| | | $C$ | $D$ | |
| $(G,G)$ | $C$ | 1 | 0 | $b>c$ |
| $(G,B)$ | $D$ | * | 1 | |
| $(B,G)$ | $C$ | 1 | 0 | |
| $(G,G)$ | $C$ | 1 | 0 | $b>2c$ |
| $(G,B)$ | $D$ | * | 1 | |
| $(B,G)$ | $D$ | * | 1 | |

Note that $S(G,G)$ must be $C$ and $S(G,B)$ must not be $C$ for the norm to be a CESS. In the following, the two actions other than $A_{GB}$ are denoted as $\{A_{GB}^{\dagger}, A_{GB}^{\dagger\dagger}\}$. For instance, if $A_{GB} = D$, then $A_{GB}^{\dagger} = C$ and $A_{GB}^{\dagger\dagger} = P$ (or vice versa). We define $A_{BG}^{\dagger}$, $A_{BG}^{\dagger\dagger}$, $A_{BB}^{\dagger}$, and $A_{BB}^{\dagger\dagger}$ similarly. The social norm is a CESS norm if and only if the following conditions are met,

$$\begin{cases}
R(G, G, C) = 1 \\
R(G, B, A_{GB}) + R(B, G, A_{BG}) > 1 \\
\left[R(G, G, C) - R(G, G, D)\right]\Delta\nu > \zeta_C - \zeta_D \\
\left[R(G, G, C) - R(G, G, P)\right]\Delta\nu > \zeta_C - \zeta_P \\
\left[R(G, B, A_{GB}) - R\left(G, B, A_{GB}^{\dagger}\right)\right]\Delta\nu > \zeta_{A_{GB}} - \zeta_{A_{GB}^{\dagger}} \\
\left[R(G, B, A_{GB}) - R\left(G, B, A_{GB}^{\dagger\dagger}\right)\right]\Delta\nu > \zeta_{A_{GB}} - \zeta_{A_{GB}^{\dagger\dagger}} \\
\left[R(B, G, A_{BG}) - R\left(B, G, A_{BG}^{\dagger}\right)\right]\Delta\nu > \zeta_{A_{BG}} - \zeta_{A_{BG}^{\dagger}} \\
\left[R(B, G, A_{BG}) - R\left(B, G, A_{BG}^{\dagger\dagger}\right)\right]\Delta\nu > \zeta_{A_{BG}} - \zeta_{A_{BG}^{\dagger\dagger}} \\
\left[R(B, B, A_{BB}) - R\left(B, B, A_{BB}^{\dagger}\right)\right]\Delta\nu > \zeta_{A_{BB}} - \zeta_{A_{BB}^{\dagger}} \\
\left[R(B, B, A_{BB}) - R\left(B, B, A_{BB}^{\dagger\dagger}\right)\right]\Delta\nu > \zeta_{A_{BB}} - \zeta_{A_{BB}^{\dagger\dagger}}
\end{cases} \quad (44)$$

Here, $\zeta_A$ is defined as the instantaneous cost of action $A$,

$$\zeta_A \equiv \begin{cases}
c & \text{if } A = C \\
0 & \text{if } A = D \\
\alpha & \text{if } A = P.
\end{cases} \quad (45)$$

The marginal long-term payoff $\Delta\nu$ is

$$\Delta\nu = \begin{cases}
b/R(B, G, C) & \text{if } (A_{GB}, A_{BG}) = (D, C) \\
(b + \beta)/R(B, G, C) & \text{if } (A_{GB}, A_{BG}) = (P, C) \\
(b - c)/R(B, G, D) & \text{if } (A_{GB}, A_{BG}) = (D, D) \\
(b - c + \beta)/R(B, G, D) & \text{if } (A_{GB}, A_{BG}) = (P, D) \\
(b - c + \alpha)/R(B, G, P) & \text{if } (A_{GB}, A_{BG}) = (D, P) \\
(b - c + \alpha + \beta)/R(B, G, P) & \text{if } (A_{GB}, A_{BG}) = (P, P).
\end{cases} \quad (46)$$

As special cases, the deterministic CESS norms with punishment, summarized in S1 Appendix, fall into six classes.

### 4.3. Leading-eight norms with non-vanishing error rate

We can also derive the ESS conditions for the leading-eight norms when the error rates are non-vanishing. Naturally, errors make the conditions for these norms to be ESS more stringent; but how does it depend on the error rates? The leading-eight norms have

$$
\begin{aligned}
&\chi_C\left(G,\Delta\right)=1 \\
&\chi_C\left(B,\Delta\right)=\begin{cases}0 & \text{for L1,L2}\\1 & \text{for L3--L8}\end{cases} \\
&\chi_C\left(\Delta,G\right)=0 \\
&\chi_C\left(\Delta,B\right)=\begin{cases}-1 & \text{for L1,L2}\\0 & \text{for L3--L8}\end{cases} \\
&R_S\left(\Delta,G\right)=0 \\
&R_S\left(\Delta,B\right)=\begin{cases}\mu_e\left(1-\epsilon_{DC}\right)\left(1-2\mu\right) & \text{for L1}\\\left(\mu_e-\epsilon_{DC}-\mu_e\epsilon_{DC}\right)\left(1-2\mu\right) & \text{for L2}\\0 & \text{for L3}\\\epsilon_{DC}\left(1-2\mu\right) & \text{for L4}\\-\epsilon_{DC}\left(1-2\mu\right) & \text{for L5}\\0 & \text{for L6}\\1-2\mu & \text{for L7}\\\left(1-\epsilon_{DC}\right)\left(1-2\mu\right) & \text{for L8}\end{cases}
\end{aligned}
\tag{47}
$$

Plugging those into Eq (15), we obtain

$$
\Delta v=\begin{cases}\dfrac{bh^*+c\left(1-h^*\right)}{1-\left(1-h^*\right)\left(1-2\mu\right)\mu_e\left(1-\epsilon_{DC}\right)} & \text{for L1}\\[2ex]\dfrac{bh^*+c\left(1-h^*\right)}{1-\left(1-h^*\right)\left(1-2\mu\right)\left(\mu_e-\epsilon_{DC}-\mu_e\epsilon_{DC}\right)} & \text{for L2}\\[2ex]b & \text{for L3}\\[1ex]\dfrac{b}{1-\left(1-h^*\right)\left(1-2\mu\right)\epsilon_{DC}} & \text{for L4}\\[2ex]\dfrac{b}{1+\left(1-h^*\right)\left(1-2\mu\right)\epsilon_{DC}} & \text{for L5}\\[2ex]b & \text{for L6}\\[1ex]\dfrac{b}{1-\left(1-h^*\right)\left(1-2\mu\right)} & \text{for L7}\\[2ex]\dfrac{b}{1-\left(1-h^*\right)\left(1-2\mu\right)\left(1-\epsilon_{DC}\right)} & \text{for L8}\end{cases}
\tag{48}
$$

Except for the second-order norms L3 and L6, these analytical expressions for $\Delta v$ contain $h^*$. While $h^*$ is analytically solvable as a root of the quadratic equation, the expression is not simple enough to provide intuition. However, for L3 and L6 we can derive a simple ESS condition based on Eq (19):

$$
\frac{b}{c}>\frac{1}{\left(1-2\mu\right)\left(1-\mu_e\right)\left(1-\epsilon_{DC}\right)}.
\tag{49}
$$

As $\mu$ increases from zero to one half, or $\mu_e$ increases from zero to one, or $\epsilon_{DC}$ increases from zero to one, the right-hand side diverges, indicating that the ESS condition becomes increasingly hard to satisfy.

Interestingly, while many previous studies concluded that L6 is the most successful norm among the leading eight in evolutionary simulations [5,12,30,31], L6 has exactly the same ESS condition as L3. This theoretical prediction is accurately reproduced in numerical calculations, as shown in Fig 2. Moreover, Eq (48) shows that the $\Delta v$ of L6 is always smaller than or equal to those of L4, L7, and L8, indicating that L6 has a smaller ESS parameter region. These results suggest that L6 is not the best norm in terms of its ESS parameter region. The evolutionary advantage of L6 over L3 cannot be explained by the size of the ESS parameter region.

Instead, the advantage of L6 over L3 may come from a larger payoff difference between residents and mutants. In Fig 3, we show the average payoff of the mutants over all possible deterministic action rules other than the residents' action rule. Since L6 has a larger payoff difference, it is better able to resist invasion by the mutants, despite having the same ESS condition as L3.

For completeness, the results for the other leading-eight norms are shown in S1 Fig 1. We compare the numerically calculated results with the theoretical predictions obtained from Eq (48), which again shows perfect agreement. According to this figure, L7 has the widest ESS region, indicating its robustness against errors.

## 4.4. Equalizer norms

Our analysis also allows us to identify a special class of norms that enforce the mutant's payoff to be the same as the payoff of the residents, irrespective of the mutant's action. We call such a

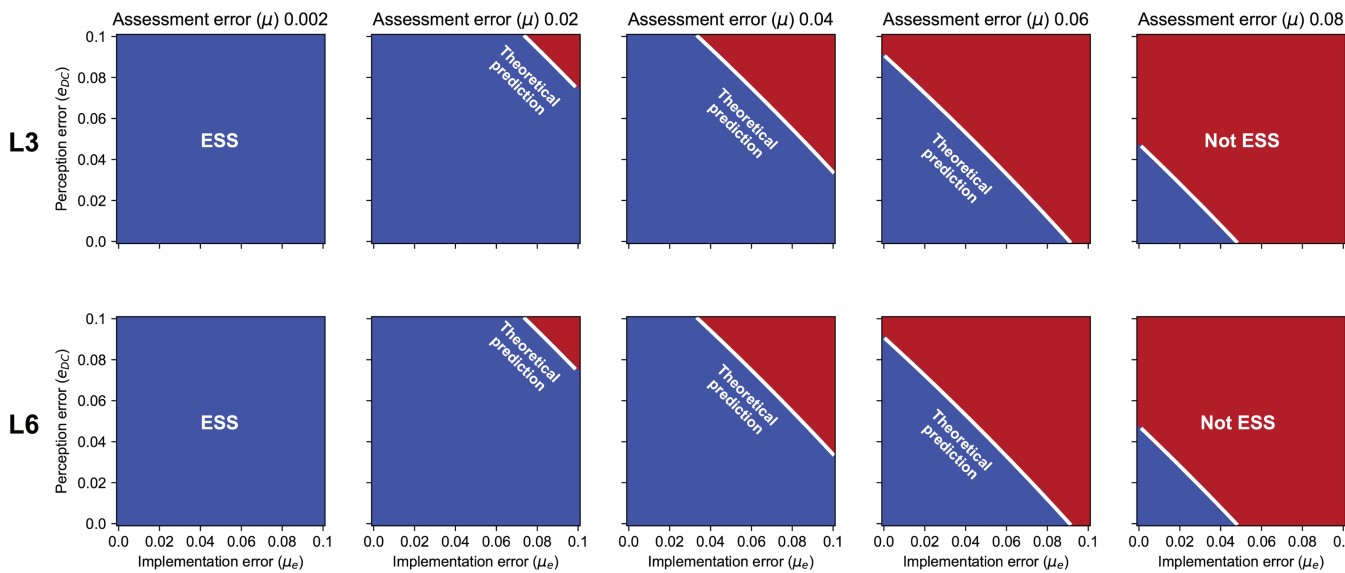

**Fig 2. ESS conditions for L3 and L6 under non-vanishing error rates.** To obtain numerical evidence, we systematically vary the assessment error rate ($\mu$), the perception error rate ($\epsilon_{DC}$), and the implementation error rate ($\mu_e$), for a game with benefit $b = 1$ and cost $c = 0.8$. The respective process is described in S1 Appendix. Regions where the ESS conditions are satisfied are shown in blue, while regions where they are not satisfied are shown in red. The solid white line represents the theoretical prediction based on Eq (23). In each case, the theoretical prediction accurately reproduces the numerical results, confirming the validity of our analysis. The figure also highlights that the ESS conditions for L3 and L6 are identical. We repeat the same analysis for the other leading-eight norms, L1, L2, L4, L5, L7, and L8. The respective results are shown in S1 Fig 1.

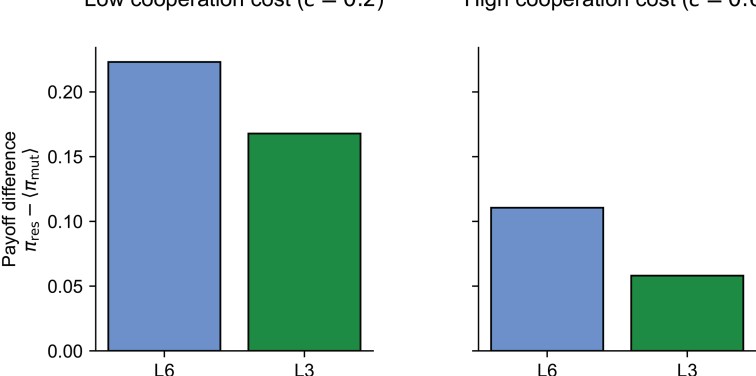

**Fig 3. Payoff differences between residents and mutants for L3 and L6 norms.** We analyze the robustness of the leading norms L3 and L6 by setting each as the resident norm and considering all 15 possible mutant deviations in the action rule. For each case, we compute the expected payoff of the resident and compare it to the average payoff of the mutants, plotting the difference. We do this for two cooperation cost scenarios: low (0.2) and high (0.6). In both scenarios, deviations from L6 result in larger payoff differences than deviations from L3, suggesting that it is more costly to deviate from L6 than from L3. Parameters: $b = 1$, and $\mu = \mu_e = \epsilon_{DC} = 0.1$.

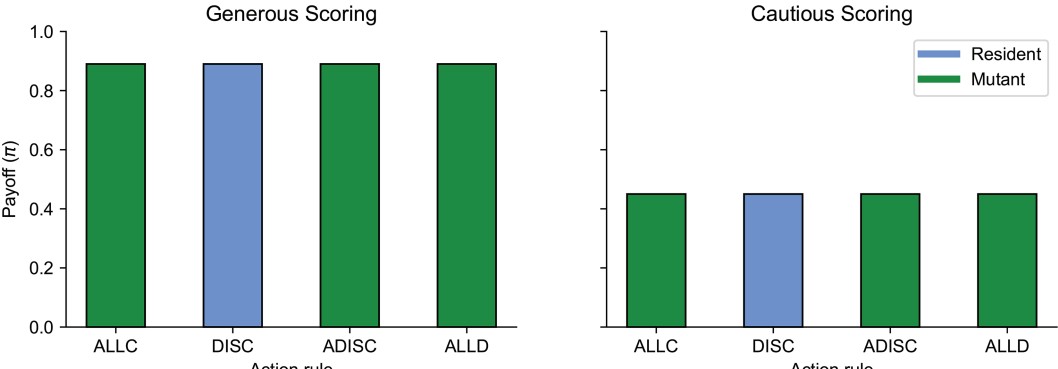

**Fig 4. Equalizer norms.** Equalizer norms impose a fixed payoff on any mutant norm. We demonstrate this property using a numerical example with two resident norms: Generous Scoring (left) and Cautious Scoring (right; defined in Eq (53)). Both norms are first-order and use the discriminator (DISC) action rule, $S(*, G) = C$ and $S(*, B) = D$. For the mutant, we consider three deterministic deviations in the action rule: ALLC (Always Cooperate: $S(*, G) = S(*, B) = C$), ALLD, and anti-discriminator (ADISC: $S(*, G) = D$, $S(*, B) = C$). As expected, the mutant's payoff equals that of the resident. Parameters: $b = 1$, $c = 0.1$, $\mu = 0.01$, and $\mu_e = \epsilon_{DC} = 0.0$.

norm an "equalizer", in analogy of the respective class of zero-determinant strategies in direct reciprocity [28].

To describe these norms formally, a social norm is an equalizer if and only if

$$\left[\widetilde{R}(X, Y, C) - \widetilde{R}(X, Y, D)\right] \Delta v = c \qquad (50)$$

holds for all possible contexts $(X, Y) \in \left\{(G, G), (G, B), (B, G), (B, B)\right\}$. When this condition holds, cooperation and defection yield identical expected payoffs. Therefore, the mutant's payoff no longer depends on the mutant's action. Such equalizer norms thus form a Nash equilibrium (but they are not an ESS since they allow for neutral invasion).

The norms described by (50) represent a generalization of the Generous
Scoring (GSCO) norm described by Schmid et al [25]. GSCO is a first-order norm defined by

$$
\begin{aligned}
S(*, G) &= C, \\
S(*, B) &= D, \\
R(*, *, C) &= 1, \\
R(*, *, D) &= 1 - \frac{c}{(1 - 2\mu)\, b}.
\end{aligned}
\tag{51}
$$

It is straightforward to show that GSCO is an equalizer. Irrespective of the applied norm of
the mutant, its payoff exactly matches the payoff of the residents.

There are other examples of equalizer norms. For example, for second-order norms with a
perfectly discriminating action rule, we have $\Delta v = b$, see Eq (16). Such a norm is an equalizer
if and only if

$$
R\left(*, Y, C\right) - R\left(*, Y, D\right) = \frac{c}{(1 - 2\mu)\, b}
\tag{52}
$$

for any $Y \in \{G, B\}$. In particular, the following is an equalizer,

$$
\begin{aligned}
S(*, G) &= C, \\
S(*, B) &= D, \\
R(*, G, C) &= 1, \\
R(*, G, D) &= 1 - \frac{c}{(1 - 2\mu)\, b}, \\
R(*, B, C) &= \frac{c}{(1 - 2\mu)\, b}, \\
R(*, B, D) &= 0.
\end{aligned}
\tag{53}
$$

To demonstrate the properties of equalizers, we present numerical examples in Fig 4. In
these examples, residents and mutants with different action rules receive exactly the same
payoffs.

## Discussion

In this paper, we focus on indirect reciprocity under public assessment. Within this setting,
we analytically characterize all third-order evolutionarily stable norms (ESS). Previously, most
studies focused on ESS that are fully cooperative when error rates were sufficiently small. Our
analysis generalizes these results to cases where the population is not fully cooperative and
errors are no longer small. In this way, we establish a more comprehensive foundation for the
theory of indirect reciprocity. This broader framework enables us to study a wider range of
social norms and to investigate their stability for arbitrary error rates. Moreover, it allows us
to explore the effects of additional actions beyond cooperation and defection – such as costly
punishment.

Based on this framework, we obtain several important insights. First, in the limit of vanish-
ing error rates and deterministic norms, our results recover the well-known leading-eight and
the secondary-sixteen norms [9,10,26]. Second, we systematically derive all cooperative ESS
for the case when a costly punishment option is available. The corresponding results success-
fully reproduce previous findings for second-order social norms [13,21]. Third, we analyze
the robustness of the leading-eight norms under varying error rates. This analysis shows that

the two second-order norms L3 and L6 have exactly the same critical benefit-to-cost ratio, even though L6 is more punitive against mutants than L3. Finally, we describe a novel class of norms, termed 'equalizers', which unilaterally fix a mutant's payoff to match that of the residents, regardless of the mutant's strategy. This is a generalization of the Generous Scoring (GSCO) norm [25] and is reminiscent of the zero-determinant strategies of direct reciprocity [28]. All of these analytical findings are further supported numerically (see also S1 Appendix).

As the main methodological innovation of our study, we focus on a key variable: the long-term benefit of having a good reputation, denoted $\Delta v$ [13,21,27]. This quantity captures the advantage of maintaining a good reputation instead of getting a bad one. It provides the critical basis for deriving necessary and sufficient conditions for all ESS, regardless of the cooperation level they sustain. In the following, we discuss how this quantity is related to previous approaches. In reinforcement learning, the value of being in a certain state, referred to as the "state value function", is calculated using the Bellman equation. Ohtsuki et al. [13] apply the Bellman equation to calculate the value of being good $v_G^{(T)}$ in the context of costly punishment (a

similar approach is used in the context of repeated games, where it is often referred to as the continuation payoff). While this method is versatile, a discount factor must be introduced to ensure that the continuation payoff converges. A simpler approach is to calculate the difference between the values of being good and bad (our $\Delta v$), which is sufficient to determine whether a norm is an ESS. Even if $v_G^{(T)}$ and $v_B^{(T)}$ both diverge, the difference $\Delta v$ remains finite, and no discount factor is needed.

In Ref. [21], the relationship $\Delta v = b$ is derived for second-order norms. This relationship is then used to calculate the ESS conditions when there is also a costly punishment option. Ref. [26] derives the ESS conditions for fully cooperative norms. There, a quantity akin to $\Delta v$ is computed assuming that the population mostly consists of good players. The present paper extends those previous analyses to general third-order norms. Our framework allows for analytical solutions, even when error rates do not vanish and when the population is not fully cooperative. Still, our analysis relies on the assumption of binary reputations. When reputations are not binary [20,32,33], analytical approaches become significantly more complex. We leave this extension for future work.

For direct reciprocity, it is possible to identify four classes of equilibrium behavior among memory-1 strategies of the repeated prisoner's dilemma [34]. In equilibrium, players are either fully cooperative, fully defective, they engage in alternating cooperation, or they apply equalizers. A natural question is whether the ESS norms of indirect reciprocity can be categorized similarly. Our analysis, however, shows that such a classification with a handful of distinct categories is infeasible. Instead, ESS norms of indirect reciprocity can support arbitrary levels of cooperation. To illustrate this point, consider a second-order norm using a discriminating action rule. The respective ESS conditions, as given by Eq (19), are $[\widetilde{R}(*,G,C) - \widetilde{R}(*,G,D)]b > c$ and $[\widetilde{R}(*,B,C) - \widetilde{R}(*,B,D)]b < c$. These inequalities constrain the differences in assessment values (e.g., $[\widetilde{R}(*,G,C) - \widetilde{R}(*,G,D)]$), but not their absolute values. As a result, a wide range of average cooperation levels can be realized in an ESS.

In our analysis, we assume that the population is monomorphic, i.e., all individuals use the same social norm, and we explore whether this norm is stable against invasion by rare mutants. While this is one of the most standard approaches to assess the stability of social norms, it is also important to consider the evolutionary dynamics of polymorphic populations, where players with multiple action rules may coexist. Furthermore, another interesting direction would be to investigate multiple social norms coexisting in a population. Although

we leave these for future work, it would be valuable to analyze the evolutionary dynamics of polymorphic populations extending the framework developed in this paper.

Finally, we note that our analysis is based on the assumption of "public assessments". That is, all individuals are assumed to agree on each others' reputations. This, of course, is a strong idealization. Many real-world social interactions may be more accurately described by a "private assessment" model, where individuals are allowed to disagree on how they view others [35–47]. Still, the public assessment model often serves as a useful reference point for theoretical exploration. Moreover, as a recent study has shown, the public assessment model and the private assessment model are not completely independent; rather, they can be unified within a single framework [35]. In light of this recent progress, we believe our analysis offers a solid foundation for advancing the understanding of indirect reciprocity, including in the context of private assessments.

## Acknowledgments

CH acknowledges generous support from the European Research Council Starting Grant 850529: E-DIRECT. YM acknowledges support by JSPS KAKENHI Grant Number JP25K07145.

## Supporting information

**S1 Appendix. S1 Appendix details the numerical verification of ESS conditions and provides a complete classification of deterministic CESS norms with punishment.** For each case, we present the norms that satisfy the CESS criteria and the corresponding parameter regimes.
(PDF)

**S1 Fig. ESS conditions of the leading eight strategies under non-vanishing error rates.** Similar to Fig 2 of the main text, we show numerical examples of the ESS conditions for the leading eight norms when error rates can be positive. Regions where the ESS conditions are satisfied are shown in blue, and those where they are not satisfied are in red. The theoretical predictions are shown as solid white line. Parameters: $b = 1$ and $c = 0.8$.
(EPS)

## Author contributions

**Conceptualization:** Yohsuke Murase.

**Formal analysis:** Nikoleta E. Glynatsi, Yohsuke Murase.

**Investigation:** Nikoleta E. Glynatsi, Yohsuke Murase.

**Methodology:** Nikoleta E. Glynatsi, Christian Hilbe, Yohsuke Murase.

**Software:** Nikoleta E. Glynatsi, Yohsuke Murase.

**Validation:** Nikoleta E. Glynatsi, Christian Hilbe, Yohsuke Murase.

**Visualization:** Nikoleta E. Glynatsi.

**Writing – original draft:** Yohsuke Murase.

**Writing – review & editing:** Nikoleta E. Glynatsi, Christian Hilbe, Yohsuke Murase.

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
