## [Decision Letter · Decision Letter 0]

8 Aug 2025

PCOMPBIOL-D-25-01270

Exact conditions for evolutionary stability in indirect reciprocity under noise

PLOS Computational Biology

Dear Dr. Glynatsi,

Thank you for submitting your manuscript to PLOS Computational Biology. After careful consideration, we feel that it has merit but does not fully meet PLOS Computational Biology's publication criteria as it currently stands. Therefore, we invite you to submit a revised version of the manuscript that addresses the points raised during the review process.

Please submit your revised manuscript within 30 days Oct 08 2025 11:59PM. If you will need more time than this to complete your revisions, please reply to this message or contact the journal office at ploscompbiol@plos.org. Please include the following items when submitting your revised manuscript:

We look forward to receiving your revised manuscript.

Kind regards,

Feng Fu

Academic Editor

PLOS Computational Biology

Zhaolei Zhang

Section Editor

PLOS Computational Biology

**Journal Requirements:**

3) We notice that your supplementary Figures, and Tables are included in the manuscript file. Please remove them and upload them with the file type 'Supporting Information'. Please ensure that each Supporting Information file has a legend listed in the manuscript after the references list.

4) Please ensure that the funders and grant numbers match between the Financial Disclosure field and the Funding Information tab in your submission form. Note that the funders must be provided in the same order in both places as well.

**Reviewers' comments:**

Reviewer's Responses to Questions

**Comments to the Authors:**

Reviewer #1: Summary of the paper

This paper provides an analytical method to determine if a given social norm, under indirect reciprocity, can theoretically sustain human cooperation in the donation game. In addition, it extends this analysis to commonly studied scenarios, such as non-vanishing error rates, and the presence of punishments. It uses these criteria to rediscover the leading eight and secondary social norms, providing a formalized understanding of why these social norms are successful.

General appreciation

The paper is written with attention and is structured in an adequate manner. It is clearly motivated as a generalised model of Murase et al., with well-justified improvements that are certainly helpful for future works of indirect reciprocity and human cooperation. While this work is largely a build up from prior works, it makes meaningful contributions and provides enough material to make it a stand on its own. In particular, prior works typically relied on simulations, or analytical approaches with simplifying assumptions which this work does not rely on. As such, it constitutes a technical advance in the field.

Major issues

Your work assumes a single strategy is present in the population, potentially invaded by a second strategy. Yet, other works in indirect reciprocity consider the potential simultaneous presence of 3 strategies: Always cooperate, always defect, and DISC. Is your analysis capable of expressing such a 3-strategy system? And how does the predicted success of each norm following your conditions differ when these 3 strategies are at play? If differences exist, they should be clarified in the discussion, or explored in an appendix. This would greatly aid the field, as many results of indirect reciprocity are incompatible and often contradictory due to their differing assumptions.

Following the point above, in Line 336 it would be a helpful addition to clarify what then differs in the evolutionary simulations and your work. That is, why is L6 commonly reported as the most successful norm? Is it a case of misreporting, or are the experiments developed in other works using different settings?

In Line 398, the connection to reinforcement learning is clear, yet under explored. Is there any insight your methods could bring to reinforcement learning, given that many new works are attempting to implement reputations to promote cooperation in multi agent systems?

Minor issues

Line 28: Clarifying what a deterministic or a stochastic norm is could aid the reader.

Line 52: Without reading ahead, it is unclear what a secondary norm is.

Line 65: While I understand the usage of “build-up a reputation over time”, the model then explicitly states that reputations are binary and players only consider the current reputations, which may be confusing to the reader.

The author summary states that social norms define how reputations change, but, from the remaining text, social norms appear to only have this effect indirectly, as they instead define what is considered good. It is better to clarify this.

Line 179: There’s a small inconsistency, it should read “second-order”.

Line 204: Although it is clear for readers familiar with indirect reciprocity, ALLD should be defined, especially S(X,Y), since otherwise it is unclear why Eq 19 is satisfied.

Line 306: From the punctuation and phrasing, the two conditions are hard to understand. In my first reading, I understood that a social norm is a CESS if either S(B, G) = C, or it satisfies Eq. 41, which itself also contains S(B, G) = C, and only in line 309 did the other condition make sense.

Line 379: Small typo at “[…] enables us to study of a wider […]”

Equation 7: It is better to clarify the superscript “res->res” for the reader, as only in the appendix is a different usage of the notation present.

Reviewer #2: I want to congratulate the authors to their very comprehensive work. I love to see the field of indirect reciprocity continue to evolve.

I also want to state, that I am familiar with the topic of indirect reciprocity, yet that I do not have a lot personal experience with the analytical approach they apply. I tried to understand the methodology as good as possible, but I cannot make a definitive statement on its soundness.

The authors work seems to be mainly based on their own previous studies [21] and [23], which tackled costly punishment (and various errors), and stochastic third-order norms respectively. In the current study, these aspects seem to be combined in one model for the first time. The authors use their model to study 4 specific cases:

1) What strategies are CESS if there are no errors – the result is a list of conditions for two classes of stochastic strategies. The subset of these which are deterministic instead reproduces the leading-eight and the secondary-sixteen norms from previous studies. This is a validation of their and the previous work.

2) Which are CESS if costly punishment is an option (apparently also if there are no errors) – the result is another list of conditions and a list of classes of deterministic third-order strategies, the second-order subset again reproducing previous work.

3) Under which condition are the leading-eight CESS under diverse errors (3 types) – The results are conditions, as well as figures that show the values of the errors for which the model and a numerical simulation show each leading-eight to be ESS. Some finer points are made about L3 and L6.

4) Description of equalizer norms (apparently again if there are only assessment errors) – This type of norm provides the same payoffs for cooperation and defection. As a result, mutants cannot earn higher payoffs than residence. The general principle is described, but conditions are only given for GSCO (another reproduced finding) and another example for which the central equation could be simplified.

In the discussion, the authors also state that their model has advantages over previous ones, the difference being the use of “Delta v” = the difference between the payoffs for a player with good reputation compared to one with bad reputation. The authors contrast this with the approach of [13].

I consider the successful reproduction of previous works is an important contribution to the field. The paper might be the first to test the stability of the leading-eight under public assessments with these specific types of errors or three different types of errors simultaneously in general. The discovery of additional CESS that use costly punishment and of other ‘zero-determinant’-like strategies are important findings as well. However, I would like to ask the authors to address some concerns about the structure of the paper, the accessibility to less familiar reader and some other points.

Especially in the beginning, but also later one, I asked myself some broader questions:

Is this paper foremost about the model, or about the results?

Why do you introduce various errors to then not use them in 3 out of 4 cases? (without commenting on it)

OR

Why were all model extension combined in a single paper, but then only applied one at a time?

I will list points I think addressing could potentially improve the paper, and I will mark some of them as [MAJOR] that would be more essential.

General issues:

The scope of the paper is quite large. The actual result sections, on the other hand, seem rather short. Some results are not even properly shown or discussed in their section (see some detailed comments below). Which makes the paper hard to grasp. It is almost as if a clear story is missing. A solution could be to split it up the paper (a rather drastic move) or move some entire section to the supplements. But it might also benefit from some simple restructuring (e.g. combine the analysis of each case with its results and a short discussion of each in a section; or to keep the topics in the analyses and cases in the same order).

[MAJOR] Probably as a consequence of the scope, that had to be packed in a few sentences, I find the abstract potentially a bit miss leading. When I first read it, I was expecting the authors would provide a list of all CESS under (various) noise. Instead, they do report all CESS under no noise, but study various noise only for the leading-eight. Also, the costly punishment and equalizer norms seems to be studied without the various noises.

Equations in general could be elaborated on. It sometimes feels like the reader is required to fully understand meanings, reasons and consequences only by the equations themselves to be able to follow the line of argument. It would help if the text provided more information to the equations and speak for itself, also making the transitions between equations easier to follow. It took me a while to understand and it is possible that I only succeeded because I read some analysis of [21] and [23], which seem easier to follow (but are also smaller in scope).

Some phrases are introduced without any explanation. Many researchers will know them, but some might not, I believe it could help to add some information about some of them e.g. Line 28 – “stochastic norms”, Line 93 – “ergodic”, “short memory effect” Line 182, “perfectly discriminating” Line 183, “ALLD” Line 204 (and it is called a norm, but which norm is it? Given all Deltas are zero, it could be all R=0 or all R=1, right? I think ALLD is most often used to describe the action rule and not the norm), “Bellman equation” line 400

Single Issues:

Over the course of reading, I have asked myself a few questions, that might be helpful indications on what could be unclear to a general audience. I don’t ask the authors to answer them all (except the ones I marked as [MAJOR]), but I wanted to provide them all in case they might be helpful.

How is equ 5 derived?

What would be a name for X in equ 8?

How can equ 15 be simplified to 16? … It took me a while to realize that the cost in 15 only occurs if a player with a bad reputation has to act differently to a player with a good reputation.

[MAJOR] Line 311: “Those two …” Does that mean that there are no stable stochastic CESS after all? And the leading-eight are not special cases but the only cases for the first and so on?

Line 331: “which is a root of the quadratic equation.” - What is the consequence of that?

What is a good summary of the additional inside in 4.1) compared to [23]?

Why is the role of L3 and L6 (and their differences) discussed in such detail, even with an additional figure 4, when Fig 5, the main finding is only in the supplements?

[MAJOR] Why is fig 4 coming after fig 3?

[MAJOR] Should it be [21] instead of [23] in line 388?

Why are the results (the CESS strategies) of 4.2) only shown in a table in the supplements?

[MAJOR] Is it not true, that table 3 has strategies that were not in previous studies (line 388)? If so, why is this not discussed?

[MAJOR] In the beginning of the discussion, the authors state that “most” other studies required error rates that are vanishingly small (which I take to mean effectively 0). Which seems a bit odd given that even (one of) the most fundamental papers [9] in 2004 already considered two types of errors >0. The statement feels a bit like an attempt to boost the importance of the current paper, a statement that I do not really agree with. If the authors had a list of important papers that required vanishingly small errors in mind, it would be helpful to cite them here.

[MAJOR] What are the “wider range of social norms, including those that tolerate moderate levels of error.” Line 379? (Or why would the norms you can look at with your extension be more able to tolerate errors, and how would you know since you only study those errors for the leading-eight?)

[MAJOR] What are the ‘leading norms’ in that context? Line 386

The use of “Delta v” is introduced as an important improvement over past models Line 397. However, with the information given in the discussion, I was not clear why that is the case. It might be my fault for not understanding the details enough. But for the sake of readers like me, the other approaches could be explained in more detail, and what makes this one better.

At last, maybe also consider the following very minor points:

Table 1 - Ohtsuki and Iwasa [9] characterized L1 as “standing” based on “Sugden, R., 1986. The Economics of Rights, Co-operation and Welfare. Blackwell, Oxford, UK.”

Line 88 – introduction of errors, which are only later specifically called “assessment errors” (It is in the sentence, but if one searches for the phrase “assessment error” they do not find this passage)

Line 108 – citation of previous use of CESS might be helpful

Line 276 – citation or explanation why they are interchangeable might be helpful as well

Again, I find the model and the findings excellent and worthwhile work. I hope the points I raised can improve the quality of the paper even more.

Reviewer #3: Summary

The manuscript develops a generalized framework to characterize analytically the evolutionary stability of social norms in indirect reciprocity under different types of errors. It extends previous work by: allowing non-vanishing error rates, incorporating stochastic norms, and accommodating additional actions beyond cooperation and defection. The paper also identifies a class of “equalizer” norms, drawing an analogy to zero-determinant strategies in direct reciprocity. The paper is well-written, the results are novel and interesting, and the mathematical treatment is sound.

Overall evaluation

The manuscript makes a substantial theoretical contribution, but to fully convey its impact it needs clearer framing of its novelty, tighter exposition in the introduction, and expanded explanation of the key methodological controbutions. These are primarily presentation and clarity issues; the underlying mathematics appears sound. So, I recommend a minor revision.

Major comments

[1] The paper would benefit from restructuring to make the novelty of the generalized framework explicit. In particular, clarify how and why this framework: (a) enables calculation of quantities that were previously more complex or intractable, (b) supports extensions such as additional actions beyond cooperation and defection, and (c) leads to the identification of equalizer norms. Bringing these points forward early will help the reader appreciate the contribution.

[2] The paragraph at lines 25–37 mostly describes Ref. [23] to set up the extension. Ref. [23] covers stochastic norms (deterministic as a special case), under public assessment, and derives ESS conditions in the limit of vanishing errors. At this point, it would be helpful to state clearly what this paper modifies or extends. The paragraph ends by hinting that the framework from [23] can be used for private assessment, but this manuscript does not address private assessment. I recommend rephrasing to avoid raising expectations that are not met. The following paragraph (lines 38–48) starts with the restriction to vanishing error rates, which has already been mentioned. Framing it as “Moreover” is confusing. Consider merging or reordering these sections to avoid repetition.

[3] The analysis considers variation only in the mutant’s action rule, not in its assessment rule. In the main text, this is assumed without explanation, and Appendix A cites Ref. [9], where it appears as a modeling postulate: “We postulate that reputation dynamics d is fixed in the population.”

While readers with expertise in the field may infer that public assessment implies a unique or external reputation mechanism that the mutant cannot influence (e.g., an external observer or a random broadcaster which very likely is the resident), this is not obvious in the paper. I recommend adding a short explanation in the main text for why only action-rule deviations are analyzed. It would also be helpful and very interesting to clarify whether the presented framework could, in principle, be adapted to include competition of assessment rules.

[4] The discussion (line 396) correctly identifies the use of \Delta v as the key methodological innovation. Because of its importance, Section 3.2 would benefit from a more detailed derivation and added intuition. The expression not only contains current payoffs that depend on the probabilities of the actions, but also future payoffs that depend on the reputation updates.

For example, lines 156–159 briefly explain the role of future payoffs through reputation updates; expanding this explanation and adding detail and intuition of the derivation would help readers understand the power of the method.

[5] In the introduction (line 42), “all evolutionarily stable social norms” is acceptable as shorthand, but in the discussion (line 374), “we analytically characterize all evolutionarily stable norms (ESS) of indirect reciprocity” feels overly strong. The analysis is limited to norms up to third order and to public assessment. Rephrasing both statements to specify this scope would be more accurate.

Minor comments

- The use of “full cooperation” needs to be clarified. Does it mean all players always choose C? In lines 39–40, what does “default state” means in the phrase “full cooperation is assumed to be the default state”?

- The statement that the principles of the leading eight norms “seem to be shared by many human communities” (lines 23–24) must be supported with a citation.

- Please specify whether “The theory” in line 34 refers to Ref. [23] or to the general theory of indirect reciprocity. With the current wording, it could be either.

- In line 411, clarify what is meant by “default reputation.” Is this the initial condition of the reputation dynamics?

- In Section 4.2, daggers are used to denote additional actions with punishment, but daggers are also used earlier for errors. Using the same symbol for both is visually confusing; consider refining the notation.

- The term “social norm” is formally defined as the combination of action and assessment rules, but when referring to the “order” of the norm it seems to mean only the assessment rule. A brief clarification of usage or consistency throughout would be helpful.

- Figure 2 lacks an x-axis label; add it for clarity even if the caption implies the content.

- Figure 4 is referenced immediately after Figure 2 and before Figure 3; and Figures 2 and 4 both compare L3 and L6. Consider swapping Figures 3 and 4 to match the order of discussion.

- Generous Scoring is described as a first-order norm in the text but labeled second-order in the caption of Figure 3. Standardize this.

- In Table 3, the triangular inequality symbols would be clearer if the terms were consistently ordered (e.g., use c > \alpha and c < \alpha rather than flipping sides) so the symbols match the inequalities.

**Have the authors made all data and (if applicable) computational code underlying the findings in their manuscript fully available?**

Reviewer #1: Yes

Reviewer #2: Yes

Reviewer #3: Yes

PLOS authors have the option to publish the peer review history of their article (what does this mean?). If published, this will include your full peer review and any attached files.

Reviewer #1: No

Reviewer #2: No

Reviewer #3: No

**Figure resubmission:**
---

## [Decision Letter · Decision Letter 1]

2 Oct 2025

Dear Dr. Glynatsi,

We are pleased to inform you that your manuscript 'Exact conditions for evolutionary stability in indirect reciprocity under noise' has been provisionally accepted for publication in PLOS Computational Biology.

Best regards,

Feng Fu

Section Editor

PLOS Computational Biology

Zhaolei Zhang

Section Editor

PLOS Computational Biology

This is an "accept", yet there are some remaining minor points as pointed out by R1. I would like to encourage the authors to take them into account as much as possible when finalizing their manuscript for publication.

Reviewer's Responses to Questions

**Comments to the Authors:**

Reviewer #1: This paper presents a mathematical framework for indirect reciprocity to assess if a social norm will theoretically sustain cooperation in the donation game. This is further extended to account for non-vanishing error rates and the presence of punishment mechanisms, recovering both the leading eight social norms and finding a class of norms analog to zero-determinant strategies.

The paper is clearly written and structured. The contributions are analytical, but clearly relevant for the field of indirect reciprocity. After presenting their methodology, a large set of scenarios are explored, providing a solid analysis for a variety of commonly studied settings.

Furthermore, the authors have integrated the changes proposed by the prior reviewers.

I will highlight the issues I found with this revision:

Although ALLD is defined, is it not the case that ALLC, DISC and ADISC are undefined? These are presented in Figure 4 and should too be explained.

In line 412, “applied norm of the mutant” suggests that mutants apply different norms, while as far as I understood only the action rule component differs. This should be clarified.

Similar to the point above, in line 163, “norm” and “action rule” are also used interchangeably. I suggest that all these instances and others I might not have found be made consistent throughout the paper.

Importantly, in line 164, should it not read “multiple assessment rules”?

In line 166 it is perhaps important to clarify what this “sometimes” is, as it is unclear how it is related to a special case, which is similarly undefined.

In addition, I suggest the following minor alterations:

In line 198, although “res” is clarified, it is left to the reader to assume that “res->res” means an interaction between two individuals using a resident strategy. This could be clarified.

There is a missing space in Line 356.

The hanging phrase at the end of Line 409 could be rephrased to be more conclusive.

I would like to emphasize that I appreciate the extensive contributions of this model and hope the points raised help clarify and strengthen the presentation of the paper.

Reviewer #2: I thank the authors for addressing all reviewer's comments. I think their efforts greatly improved the paper.

As a very small editing remark, in the paragraph of line 134, it reads as if cooperative ESS are introduced the first time in the text, but they are already mentioned in line 15.

I want to congratulate the authors again for their comprehensive work and recommend that the revised manuscript will be accepted for PLOS.

Reviewer #3: Summary of revision:

The authors have successfully addressed all my previous comments. The introduction has been significantly strengthened, ambiguities have been resolved, and the structure and presentation are now clear.

Overall evaluation:

I did not find any major issues with the current version. I describe below only a minor point. Therefore, I recommend acceptance after addressing the minor comment.

Minor comment:

- At Line 60, I suggest revising for clarity and completeness. The derived conditions are not only for “various error rates,” but for various types of errors at arbitrary rates.

I thank the authors for carefully addressing the feedback and congratulate them on their work.

**Have the authors made all data and (if applicable) computational code underlying the findings in their manuscript fully available?**

Reviewer #1: Yes

Reviewer #2: Yes

Reviewer #3: Yes

PLOS authors have the option to publish the peer review history of their article (what does this mean?). If published, this will include your full peer review and any attached files.

Reviewer #1: No

Reviewer #2: No

Reviewer #3: No

---

## [Editor Report · Acceptance letter]

PCOMPBIOL-D-25-01270R1

Exact conditions for evolutionary stability in indirect reciprocity under noise

Dear Dr Glynatsi,

I am pleased to inform you that your manuscript has been formally accepted for publication in PLOS Computational Biology. Your manuscript is now with our production department and you will be notified of the publication date in due course.

With kind regards,

Anita Estes
